# Antibiotic Resistance and Molecular Characterization of *Cronobacter sakazakii* Strains Isolated from Powdered Infant Formula Milk

**DOI:** 10.3390/foods11081093

**Published:** 2022-04-11

**Authors:** Babak Pakbin, Wolfram Manuel Brück, Samaneh Allahyari, John W. A. Rossen, Razzagh Mahmoudi

**Affiliations:** 1Institute for Life Technologies, University of Applied Sciences Western Switzerland Valais-Wallis, 1950 Sion, Switzerland; wolfram.bruck@hevs.ch; 2Medical Microbiology Research Center, Qazvin University of Medical Sciences, Qazvin 15315-3419, Iran; samaneh.alahyari66@gmail.com (S.A.); r.mahmodi@yahoo.com (R.M.); 3Department of Medical Microbiology and Infection Prevention, University Medical Center Groningen, University of Groningen, 9713 GZ Groningen, The Netherlands; j.w.a.rossen@rug.nl; 4Department of Pathology, University of Utah School of Medicine, 30 N 1900 E, Salt Lake City, UT 84132, USA; 5Laboratory of Clinical Microbiology and Infectious Diseases, Isala Hospital, 8025 AB Zwolle, The Netherlands

**Keywords:** *Cronobacter sakazakii*, powdered infant formula milk, antibiotic resistance, ERIC-PCR

## Abstract

Background: *Cronobacter sakazakii* is a new emerging foodborne bacterial pathogen associated with severe lethal diseases such as meningitis, necrotizing enterocolitis, and septicemia in infants and neonates. Powdered infant formula milk (PIFM) has been recognized as one of the main transmission vehicles and contaminated sources of this pathogen. This study aimed to investigate the prevalence rate, genotypic and phenotypic antibiotic resistance profile, and clonal relatedness of *C. sakazakii* strains isolated from 364 PIFM samples collected from Tehran city, Iran. Methods: Culture-based methods, Kirby–Bauer disk diffusion antibiotic resistance testing, conventional Polymerase Chain Reaction (PCR), and Enterobacterial Repetitive Intergenic Consensus PCR (ERIC-PCR) assays were used in this study to detect and characterize the *C. sakazakii* isolates. Results: We isolated 25 *C. sakazakii* strains from PIFM samples (6.86%). The isolates were highly resistant to amoxicillin-clavulanic acid, amoxicillin, ampicillin, cefoxitin, cefepime, erythromycin, ceftriaxone, ciprofloxacin, and chloramphenicol and susceptible to gentamicin, tetracycline, norfloxacin, and azithromycin antibiotics. The blaCTX-M-1 gene was detected in 96% of the isolates. The isolates were categorized into eight distinct clonal types using the ERIC-PCR method, showing a high genetic diversity among the isolates. However, there was a significant correlation between the genotypic and phenotypic antibiotic resistance properties of the isolates. Conclusions: Novel microbial surveillance systems for detecting multi-drug-resistant *C. sakazakii* are required to control the contamination of this foodborne pathogen in infant foods.

## 1. Introduction

*Cronobacter sakazakii*, formerly known as *Enterobacter sakazakii*, is a Gram-negative, non-spore-forming, motile, peritrichous rod emerging bacterial foodborne pathogen belonging to the Enterobacteriaceae family [1]. *C. sakazakii* is an opportunistic pathogen causing several outbreaks worldwide and may cause severe infant meningitis, necrotizing enterocolitis, and septicemia with lethality rates between 40 and 80%. Additionally, this pathogen has been associated with several cases of aspiration pneumonia, urinary tract infections, abscess, wounds, diarrhea, and conjunctivitis in adults [2]. Elderly and newborns are the population groups most affected by this pathogen. However, *C. sakazakii* mainly infects neonates under one year of age [3]. *C. sakazakii*, as a foodborne pathogen, has been isolated from different food categories, including ready-to-eat foods, retail foods, dry powdered rice, and powdered infant formula milk (PIFM). Consumption of PIFM contaminated with *C. sakazakii* is the main reason for the incidence of intestinal and extra-intestinal infections caused by this pathogen in neonates [4].

During the first months of life, human and infant formula milk play a crucial role in the nutrition of neonate and infant. Human milk has been strongly recommended by international health and nutritional organizations and is recognized as the infant feeding gold standard. Additionally, PIFM (recognized as the best alternative) is widely used in the case of human milk being unavailable [5]. PIFM, the primary contamination source of *C. sakazakii*, is mainly due to contaminated manufacturing plants or ingredients used during production [6]. The incidence rate of *C. sakazakii* in PIFM products ranges from 3 to 30% worldwide. It is worth noting that due to the reconstitution of PIFM with slightly warm water, the viability of this pathogen may not be affected [7]. Consequently, consuming contaminated PIFM poses a serious risk of lethal meningitis and enteric infections, especially for immunodeficient and immature neonates.

Antibiotic therapy is the preferred way to prevent and treat *C. sakazakii* infections in neonates [8]. However, antibiotic resistance in bacterial foodborne pathogens is a significant challenge associated with high mortality and morbidity of foodborne infectious diseases resulting in high socioeconomic costs [9]. *C. sakazakii* is usually susceptible to the most generally clinically used antibiotics. However, resistance to one or more old-generation antimicrobial agents such as streptomycin, tetracycline, cephalothin, and gentamicin has been reported [10]. Notably, infections caused by this pathogen, especially those resulting in septicaemia and meningitis, require effective antibiotic treatments [11]. Therefore, resistant *C. sakazakii* strains in infant products are a serious problem, also as the consumers are generally immunologically vulnerable. Careless use of different antibiotics in animal farms and agriculture has encouraged multiple antibiotic resistance in *C. sakazakii* isolates [12]. 

Genotyping methods including Polymerase Chain Reaction (PCR) (DNA fingerprinting) and sequencing-based assays such as multi-locus sequence typing, random amplification of polymorphic DNA, repetitive element sequence-based PCR, ribotyping, and enterobacterial repetitive intergenic consensus (ERIC) PCR methods have been considered as practical tools to perform epidemiological surveillance to establish the similarity of bacterial pathogens isolated from food, environmental, and clinical samples [13,14]. Enterobacterial Repetitive Intergenic Consensus PCR (ERIC-PCR) is a rapid, cost-effective, and relatively efficient DNA fingerprinting PCR (ERIC sequences) method for genotyping Enterobacteriaceae family strains [15]. Several researchers used this method to analyze the genetic relatedness of *C. sakazakii* strains, and they found it efficient and practical in microbiological and epidemiological studies [16]. As mentioned, *C. sakazakii* is an emerging multi-drug-resistant genetically diverse foodborne pathogen commonly isolated from infant formula, consumed by neonates [12]. Thus, a better understanding of antibiotic resistance properties and genetic relatedness of *C. sakazakii* isolates from PIFM samples could help us to prevent the infections caused by this pathogen in neonates more efficiently. Although some previous studies regarding the prevalence, ecology, and antimicrobial susceptibility of *Cronobacter* spp. were conducted in Iran and several other countries around the world [8,9,10,11,12], data regarding the molecular characterization and antibiotic resistance of this foodborne pathogen are still limited in Iran. The present study characterized antibiotic resistance phenotypes, genes, and genetic relatedness of *C. sakazakii* strains isolated from industrial PIFM samples collected from local grocery stores in Tehran, Iran.

## 2. Materials and Methods

### 2.1. Collection of PIFM Samples

A total of 364 PIFM samples (according to the sample size formula [12], including different market brands commonly consumed in Iran) were purchased and collected from 46 local grocery stores located in different areas throughout Tehran city, Iran, between July 2016 to September 2019. All samples were transported to the food microbiology laboratory and stored in their original package at a cool and dry place before any microbiological analysis for *C. sakazakii* isolation.

### 2.2. Isolation and Identification of C. sakazakii

*C. sakazakii* was isolated from the collected PIFM samples and identified using the previously described method by Fei et al. (2017) [17]. Under aseptic conditions, twenty-five grams of each homogenized PMIF sample was dissolved and pre-enriched selectively with 225 mL of prewarmed sterilized buffered peptone water (BPW, HiMedia Laboratories, Mumbai, India) following incubation for 24 h at 37 °C. A measure of 10 mL of pre-enriched sample was mixed well with 90 mL of Enterobacteriaceae Enrichment broth (EE-broth, HiMedia Laboratories, Mumbai, India) at 44 °C for 18 h. Presumptive *C. sakazakii* strains were isolated from the inoculated enrichment media through surface plating on Violet Red Bile Glucose Agar (VRBGA, HiMedia Laboratories, Mumbai, India) at 44 °C for 24 h. The purple-red colonies, surrounded by colorless zone formation, were selected and subjected to an API 20E identification system (BioMerieux, Marcy-l’Étoile, France) and biochemically confirmed. *C. sakazakii* type strain ATCC 29544 was used as the positive control and reference strain [17].

### 2.3. Phenotypic and Genotypic Antimicrobial Susceptibility Profile of C. sakazakii Isolates

Antibiotic resistance testing of *C. sakazakii* isolates was carried out using the Kirby–Bauer disk diffusion method [18]. Twenty commercial antibiotic disks (Oxoid, UK) including nalidixic acid (NAL), 30 µg; cefoxitin (FOX), 30 µg; azithromycin (AZM), 15 µg; streptomycin (SPT), 10 µg; amoxicillin-clavulanic acid (AMC), 20/10 µg; imipenem (IPM), 10 µg; kanamycin (KAN), 30 µg; amoxicillin (AMX), 25 µg; norfloxacin (NOR), 10 µg; ampicillin (AMP), 10 µg; tetracycline (TET), 30 µg; levofloxacin (LVX), 5 µg; cefepime (FEP), 30 µg; gentamicin (GEN), 10 µg; colistin (CST) 10 µg; chloramphenicol (CHL), 30 µg; trimethoprim-sulfamethoxazole (SXT), 1.25/23.75 µg; erythromycin (ERY), 15 µg; ceftriaxone (CTR), 30 µg and ciprofloxacin (CIP), 5 µg were used in this study. The antibiotic resistance patterns were interpreted according to the CLSI guidelines [18]. The presence of beta-lactamase resistance genes including *bla*_TEM_, *bla*_OXA_, *bla*_SHV_, *bla*_CTX-M-1_, *bla*_CTX-M-2_, *bla*_CTX-M-8_, and *bla*_CTX-M-9_ in the isolates was researched by conventional PCR assays using the specific primers and thermal cycling programs previously described by Dallenne et al. (2010) [19] to reveal the genotypic antibiotic resistance profile of the *C. sakazakii* isolates. *Escherichia coli* ATCC 25922, *Klebsiella pneumoniae* ATCC 700603, and *Staphylococcus aureus* ATCC 25923 were used as positive and negative controls.

### 2.4. DNA Extraction

All *C. sakazakii* and control strains were inoculated and grown in bovine heart infusion (BHI, HiMedia Laboratories, Mumbai, India) broth overnight at 37 °C. Bacterial strains were subjected to total DNA extraction using the Sinaclon Gram-negative bacterial DNA extraction kit (Sinaclon Co., Tehran, Iran) according to the manufacturer’s instruction. The quality and quantity of the extracted genomes were evaluated using a NanoDrop-1000 spectrophotometer (Thermo Scientific, Waltham, MA, USA). The concentrations of all extracted genomes were adjusted to 50 μg/mL with phosphate-buffered saline (PBS, HiMedia Laboratories, Mumbai, India) prior to the PCR reactions.

### 2.5. ERIC-PRC Genotyping

An ERIC-PCR method was carried out for genotyping *C. sakazakii* isolates according to the method previously described by Ye et al. (2010) using the two primers ERIC1R 5′-ATG TAA GCT CCT GGG GAT TCA C-3′ and ERIC2 5′-AAG TAA GTG ACT GGG GTG AGC G-3′ [20]. For each strain, ERIC-PCR was performed in a 20 µL reaction volume containing 10 µL of 2X conventional PCR master mix (Ampliqon, Odense, Denmark), 1 µL of DNA template (50 ng/µL), 1 µL of each primer (10 pM), and nuclease-free water up to the final reaction volume. The thermal cycling program was: 5 min initial denaturation at 95 °C followed by 35 cycles of 1 min denaturation at 95 °C, 1 min annealing at 52 °C, and 8 min elongation at 65 °C. The amplification products were subjected to gel electrophoresis in 1.0% (*w/v*) agarose gel for 1.5 h at 100 V and then documented using UV transillumination and gel documentation systems (NovinPars Co., Tehran, Iran). ERIC-PCR profiles were analyzed using the PyElph software [21]. The dendrogram was constructed using the Dice coefficient and the Unweighted Pair Group Method with Arithmetic averages (UPGMA) method by NTSYS-pc software version 2.11 [22]. The ERIC-PCR pattern types of *C. sakazakii* isolates with a high similarity index (≥0.5) were regarded as closely related ERIC-PCR pattern types.

### 2.6. Statistical Analysis

The Chi-square test was employed for the evaluation of significant differences (*p* < 0.05) between the prevalence rates using the SPSS software version 22.0.1 (Chicago, IL, USA). All measurements were carried out in triplicates.

## 3. Results

### 3.1. Prevalence of C. sakazakii

In this study, we isolated *C. sakazakii* in 25 (6.86%) out of 364 PIFM samples, all confirmed by the API-20E biochemical evaluation.

### 3.2. Antibiotic Resistance Phenotypes and Genotypes in C. sakazakii Isolates

Antibiotic resistance against nine different antibiotic categories and twenty commercial antibiotics was determined for all *C. sakazakii* isolates. Figure 1 shows the phenotypic resistance of *C. sakazakii* isolates against different antibiotics. All isolates were susceptible to trimethoprim/sulfamethoxazole and levofloxacin antibiotics. In contrast, *C. sakazakii* isolates were highly resistant to amoxicillin-clavulanic acid (24 isolates; 96%), amoxicillin (24 isolates; 96%), ampicillin (24 isolates; 96%), cefoxitin (23 isolates; 92%), cefepime (23 isolates; 92%), erythromycin (23 isolates; 92%), ceftriaxone (20 isolate; 80%), ciprofloxacin (14 isolates; 56%) and chloramphenicol (13 isolates; 52%) antibiotics. Also, these isolates were highly susceptible to gentamicin (1 isolate; 4%), tetracycline (1 isolate; 4%), norfloxacin (1 isolate; 4%) and azithromycin (2 isolates; 8%) antibiotics. As shown in Table 1, 24 (96%) out of 25 *C. sakazakii* isolates were resistant to at least three different antibiotic classes and considered multi-drug resistant (MDR). The multi-drug resistance pattern, including resistance to β-Lactam, Quinolone-fluoroquinolone, and Macrolide antibiotic classes, was the most prevalent MDR profile among the isolates (11 isolates; 44%). However, 8 (32%) out of 25 *C. sakazakii* isolates were resistant to an MDR profile consisting of six different classes of antibiotics: β-Lactam, Aminoglycoside, Lipopeptide, Phenicol, Quinolone-fluoroquinolone, and Macrolide. Several β-Lactamase encoding genes were detected in the isolates: *bla*_CTX-M-1_ (*n* = 24, 96%), *bla*_TEM_ (*n* = 3, 12%), *bla*_CTX-M-8_ (*n* = 3, 12%), *bla*_CTX-M-2_ (*n* = 5, 20%), *bla*_SHV_ (*n* = 6, 24%), *bla*_OXA_ (*n* = 7, 28%) and *bla*_CTX-M-9_ (*n* = 8, 32%) (Figure 2).

### 3.3. Genotyping of C. sakazakii Isolates

This study also evaluated genotypic polymorphism and genetic diversity among the *C. sakazakii* strains using ERIC-PCR fingerprint pattern analysis. Each *C. sakazakii* isolate produced ERIC-PCR banding patterns of 9 to 17 amplified PCR products, ranging from 1000 to 1800 bp (Figure 3). The *C. sakazakii* strains formed eight major clusters (E1–E8) with at least 50% ERIC-PCR profile similarity (cut-off value at 0.5). The Simpson’s Diversity Index was calculated at 0.83, demonstrating a high level of genetic diversity among the isolates. Table 2 shows the resistance phenotype, genotype, and ERIC-PCR types of the *C. sakazakii* isolates. Eight (32%) isolates were grouped in cluster E4. Clusters E1 and E3 included five isolates; however, other clusters contained only one (E2, E5, and E7) or two isolates (E6 and E8). As shown in Table 2, more antibiotic resistance genes were detected in *C. sakazakii* isolates with higher phenotypic antibiotic resistance. However, no significant relationship was observed between the ERIC-PCR clustering patterns and the phenotypic/genotypic antibiotic resistance profiles.

## 4. Discussion

*C. sakazakii* has recently been regarded as an emerging foodborne opportunistic pathogen isolated commonly from low-moisture foods, including powdered infant milk and formula [23]. *C. sakazakii* usually causes severe morbidity through some infectious diseases including meningitis, sepsis, necrotizing enterocolitis, and urinary tract infections in neonates and infants; however, this pathogen also significantly affects other age groups and leads to health complications in adults [24]. PIFM has recently been regarded the main vehicle of *C. sakazakii* transmission to infants implicating neonatal infections and has been considered as one of the most important concerns in children health around the world [6]. Other sources of this pathogen were also involved in cases of infectious diseases in children and other sensitive groups of people [25]. So far, limited studies are available on the prevalence rate of *C. sakazakii* PIFM. 

In the present study, we determined the prevalence rate of this pathogen in PIFM samples collected from different grocery stores located in Tehran city, Iran. 6.86% of PIFM samples were contaminated with *C. sakazakii*. The prevalence rate of *C. sakazakii* in this study (Iran) was higher than that reported from Ireland (3.40%; 16 out of 470 PIF samples) [26] and China in 2016 (2.77%; 56 out of 2020 PIFM samples) [17], and lower than that reported from China in 2012 (23.0%; 84 out of 366 PIFM and baby food samples) [27], the USA (26.9%; 21 out of 78 households) [28], and Korea (83.0%; 113 out of 136 infant food samples) [29]. The prevalence rate in Tehran (6.86%) does not significantly differ from that in Qazvin city, Iran (5.08%; *p* < 0.05), reported in our previous study [12]. Contamination with *C. sakazakii* usually occurs during the post-pasteurization packaging or adding ingredients to the infant formula foods [23]. The thermal resistance of this pathogen is significantly higher than that of other *Enterobacteriaceae* species [30]. Stabilized membrane proteins and phospholipids and trehalose production and accumulation by *C. sakazakii* protect this pathogen from the dry conditions in low-moisture foods such as infant formula and improve its thermal resistance. Moreover, high solid sugar and fat content in infant formula foods also protect *C. sakazakii* from thermal stress [25,31,32]. Consequently, *C. sakazakii* can survive when exposed to hot and dry conditions. Our surveillance results in this and previous studies showed serious levels of *C. sakazakii* contamination in PIFM. Also, foodborne pathogens from contaminated foods may be introduced to other processed foods and food preparation areas as an important vehicle of cross-contamination. Depending on the initial solving water temperature, *C. sakzakii* may be removed during the infant food preparation [28]. 

The emergence of MDR foodborne bacterial pathogens has recently been regarded as one of the most important concerns in public health and a major challenge in food safety. Since a wide range of clinical and veterinary antibiotics are still being used worldwide for diseases prevention, growth promotion, and treatment of sick farm animals, the prevalence of MDR foodborne pathogens has been increasing [33]. It is worthwhile to note that antibiotic resistance genes can be transferred horizontally among different species of Enterobacteriaceae family isolates in humans, animals, environments, and foods [34]. In this study, we isolated 25 *C. sakazakii* strains from PIFM samples, collected from Tehran city (Iran), mostly resistant to amoxicillin-clavulanic acid, amoxicillin, ampicillin, cefoxitin, cefepime, erythromycin, and ceftriaxone, and utterly susceptible to trimethoprim/sulfamethoxazole and levofloxacin antibiotics. A previous study in Korea by Kim et al. (2008) showed a high level of antibiotic susceptibility among the *C. sakazakii* strains isolated from infant foods against ciprofloxacin, tetracycline, chloramphenicol, kanamycin, gentamicin, and nalidixic acid antibiotics; they reported 31.8 and 5.3% of the isolates resistant to ampicillin and cephalothin antibiotics, respectively [29]. A study carried out in the United States by Kilonzo-Nthenge et al. (2012) reported that 76.1, 66.6, 57.1, and 47.6% of *C. sakazakii* isolates were resistant to penicillin, tetracycline, ciprofloxacin, and nalidixic acid antibiotics, respectively, and all isolates were susceptible to gentamicin [28]. Another study conducted in China by Li et al. (2016) showed high resistance to amoxicillin-clavulanic acid, rifampicin, tetracycline, streptomycin, and ampicillin antibiotics among *C. sakazakii* isolated from retail milk-based infant foods [27]. Moreover, in China, Fei et al. (2017) isolated *C. sakazakii* from PIF samples with the highest resistance to cephalothin antibiotics [17]. Recently, we (in the year 2020) also performed a study in Iran, Qazvin city, and showed that *C. sakazakii* isolated from infant food samples were resistant to ampicillin, amoxicillin, ciprofloxacin, and tetracycline and susceptible to chloramphenicol, amikacin, and levofloxacin antibiotics [12]. Interestingly, the antibiotic resistance pattern of *C. sakazakii* isolated from infant food samples in this study is in accordance with other studies. However, unsupervised and reckless use of different clinical and veterinary antibiotics for improving animal health worldwide may contribute to some differences observed among the antibiotic resistance patterns of *C. sakazakii* strains isolated in different studies/geographic areas [17,35]. 

In this study, 96% of *C. sakazakii* isolates were MDR, which is higher than that previously reported in the USA and China [17,28]. The *bla*_CTX-M-1_ antibiotic resistance gene was mostly detected in our isolates. Farm animals exposed to different clinical and veterinary antibiotics for a long time can develop bacterial flora harboring different antibiotic resistance genes and resistance to various antibiotics [36]. *bla*_CTX-M-1_ is one of the most prevalent antibiotic resistance genes encoding extended-spectrum beta-lactamase (ESBL) in the Enterobacteriaceae family. Bacterial strains harboring *bla*_CTX_ genes are commonly resistant to cephalosporin antibiotics such as cefoxitin, cefepime, ceftriaxone, etc. [37,38]. Notably, *C. sakazakii* strains isolated in this study harbored *bla*_CTX_ genes and were resistant to cefoxitin, cefepime, and ceftriaxone antibiotics. 

Different PCR-based genomic fingerprinting methods such as RAPD, ERIC, BOX, and rep-PCR assays have been used to evaluate the clonal relatedness and genetic diversity among foodborne pathogens isolated from food, environmental and clinical samples [13]. In this study, we used the ERIC-PCR patterns to determine the genetic relationship among the *C. sakazakii* strains isolated from the PIFM samples [20]. The results of ERIC-PCR indicated that eight (32%) isolates clustered together (E4). In addition, there were two clusters (E1 and E3), each containing five isolates and five clonal groups (E2, E5-E8), each containing just one or two isolates. No significant association between the antibiotic resistance and ERIC-PCR patterns of *C. sakazakii* isolates was found [20,39]. This contrasts with a study of Ye et al. (2008) in which they found a significant correlation between the antibiotic resistance profiles and genomic characterization of *C. sakazakii* isolates from instant formula milk [20]. However, significant positive correlations between the genotypic and phenotypic properties of antibiotic resistance were observed in most isolates. More investigations are needed to study the correlation between the genomic characterization and antimicrobial susceptibility profiles of *C. sakazakii* strains isolated from infant food samples using novel genomic analysis techniques.

## 5. Conclusions

In conclusion, we investigated the prevalence rate, antibiotic resistance, and genetic diversity of *C. sakazakii* strains isolated from PIFM samples collected from Tehran, Iran. *C. sakazakii* contamination in PIFM samples was frequently observed. This study showed that *C. sakazakii* isolates were resistant to amoxicillin-clavulanic acid, amoxicillin, ampicillin, cefoxitin, cefepime, erythromycin, and ceftriaxone antibiotics. The beta-lactamase encoding *bla*_CTX-M-1_ gene was detected in most isolates, and the isolates harboring this gene were resistant to cephalosporin antibiotics, including cefoxitin, cefepime, and ceftriaxone. Additionally, a high level of genetic diversity was observed among the isolates. We found a significant correlation between the phenotypic and genotypic antibiotic resistance properties of *C. sakazakii* isolates. Careless and irrational use of veterinary and clinical antimicrobial agents in farm and domestic animals may lead to the emergence of MDR *C. sakazakii* in infant foods.

## Figures and Tables

**Figure 1 foods-11-01093-f001:**
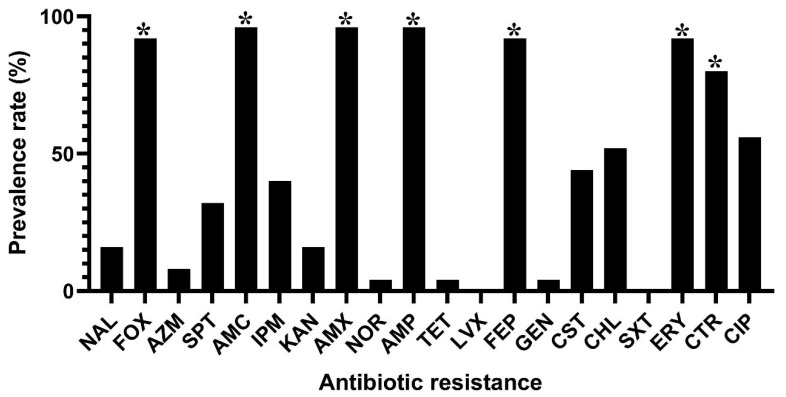
Prevalence rates of different antibiotic resistance phenotypes in *C. sakazakii* strains isolated from PIFM samples. * indicates significant differences (*p* > 0.05). NAL, nalidixic acid; FOX, cefoxitin; AZM, azithromycin; SPT, streptomycin; AMC, amoxicillin-clavulanic acid; IPM, imipenem; KAN, kanamycin; AMX, amoxicillin; NOR, norfloxacin; AMP, ampicillin; TET, tetracycline; FEP, cefepime; GEN, gentamicin; CST, colistin; CHL, chloramphenicol; ERY, erythromycin; CTR, ceftriaxone and CIP, ciprofloxacin.

**Figure 2 foods-11-01093-f002:**
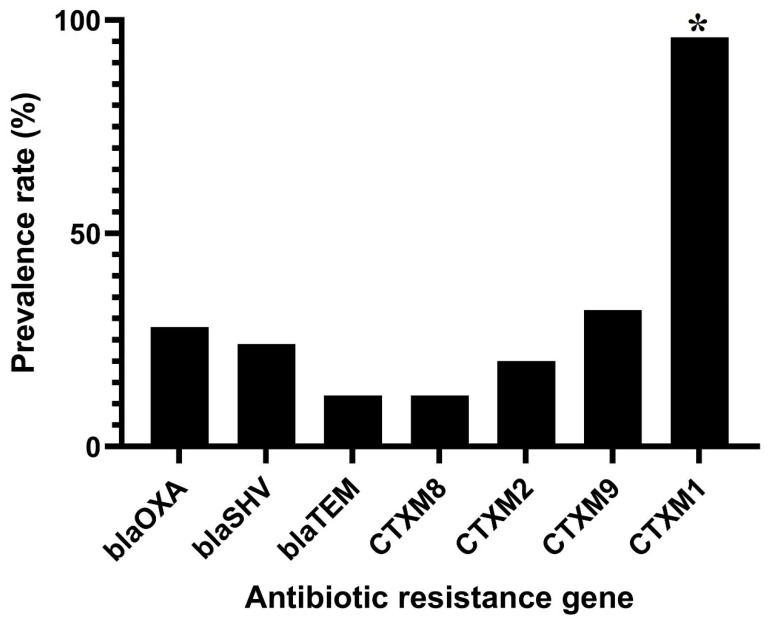
Prevalence rates of ESBLs genes in *C. sakazakii* strains isolated from PIFM samples. * indicates significant differences (*p* > 0.05). blaSHV, *bla*_SHV_; blaOXA, *bla*_OXA_; blaTEM, *bla*_TEM_; CTXM1, *bla*_CTX-M-1_; CTXM2, *bla*_CTX-M-2_; CTXM8, *bla*_CTX-M-8_ and CTXM9, *bla*_CTX-M-9_.

**Figure 3 foods-11-01093-f003:**
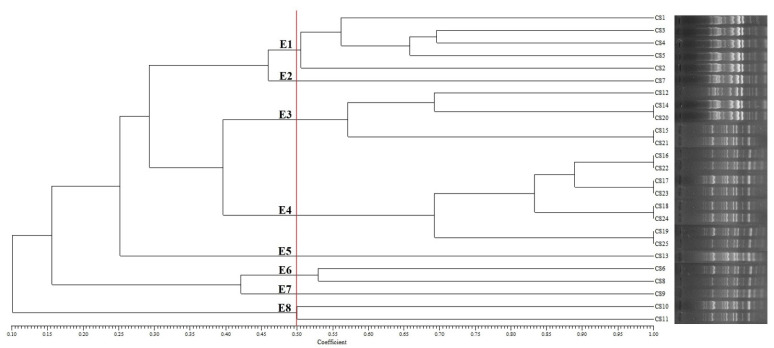
UPGMA dendrogram of *C. sakazakii* isolates from PIFM samples indicating the Dice coefficient based on ERIC-PCR analysis.

**Table 1 foods-11-01093-t001:** Multidrug resistance class patterns of *C. sakazakii* strains isolated from PIFM samples.

No. Classes of Antibiotics	Multidrug Resistance Patterns ^a^ (No. Isolates in Each Pattern)	No. Total Isolates (%) (*n* = 25)
Two	βLs-QNs (*n* = 1)	1 (4)
Three	βLs-QNs-MLs (*n* = 7)	11 (44)
	βLs-PNs-MLs (*n* = 4)	
Four	βLs-AGs-LPs-MLs (*n* = 1)	3 (12)
	βLs-PNs-MLs-QNs (*n* = 1)	
	βLs-AGs-LPs-QNs (*n* = 1)	
Five	βLs-QNs-MLs-AGs-TCs (*n* = 1)	2 (8)
	βLs-QNs-MLs-AGs-LPs (*n* = 1)	
Six	βLs-QNs-AGs-LPs-MLs-PNs (*n* = 8)	8 (32)

^a^ βLs, β-lactams; AGs, aminoglycosides; LPs, lipopeptides; TCs, tetracyclines; PNs, phenicols; QNs, quinolones and fluoroquinolones; MLs, macrolides.

**Table 2 foods-11-01093-t002:** Resistance phenotype, genotype and ERIC-PCR types in *C. sakazakii* strains isolated from PIFM samples.

No.	Isolate	Resistance Phenotype ^a^	Resistance Genotype ^b^	ERIC-PCR Type
1	CS1	NAL, FOX, AZM, AMC, IPM, KAN, AMX, AMP, TET, FEP, GEN, ERY	C1, S, O, T	E1
2	CS2	NAL, FOX, AMC, IPM, KAN, AMX, AMP, FEP, ERY, CST	C1, C2, C9	E1
3	CS3	FOX, AZM, AMC, IPM, KAN, AMX, AMP, FEP, ERY, CST	C1, S, O	E1
4	CS4	FOX, AMC, IPM, KAN, NOR, AMP, FEP, CST	C1	E1
5	CS5	NAL, FOX, AMC, AMX, AMP, FEP, ERY, CTR	C1	E1
6	CS6	FOX, AMC, AMX, AMP, FEP, CHL, ERY, CTR	C1	E6
7	CS7	FOX, SPT, AMC, AMX, AMP, FEP, CST, CHL, ERY, CTR, CIP	C1, S, O, T	E2
8	CS8	FOX, AMC, IPM, AMX, AMP, FEP, ERY, CTR, CIP	C1	E6
9	CS9	FOX, AMC, AMX, AMP, FEP, CHL, ERY, CTR	C1	E7
10	CS10	SPT, AMC, AMX, AMP, FEP, CST, CHL, ERY, CTR, CIP	C1, C8, C9	E8
11	CS11	FOX, AMC, IPM, AMX, AMP, FEP, ERY, CTR, CIP	C1, C2, C9	E8
12	CS12	FOX, SPT, AMC, AMX, AMP, CST, CHL, ERY, CTR, CIP	C1, S, O	E3
13	CS13	FOX, AMC, IPM, AMX, AMP, FEP, ERY, CIP	C1	E5
14	CS14	FOX, AMC, AMX, AMP, FEP, CHL, ERY, CIP	C1	E3
15	CS15	FOX, AMC, IPM, AMX, AMP, FEP, CST, CHL, ERY, CTR, CIP	C1, C8, C9	E3
16	CS16	NAL, FOX, AMC, AMX, AMP, FEP, ERY, CTR	C1	E4
17	CS17	FOX, AMC, AMX, FEP, CHL, ERY, CTR,	C1	E4
18	CS18	FOX, SPT, AMC, AMX, AMP, FEP, CST, CHL, ERY, CTR, CIP	C1, C2, C9	E4
19	CS19	IPM, AMX, AMP, FEP, CTR, CIP	C1	E4
20	CS20	FOX, AMC, AMX, AMP, FEP, CHL, ERY, CTR	C1, S, O	E3
21	CS21	FOX, SPT, AMC, AMX, AMP, CST, CHL, ERY, CTR, CIP	C1, S, O, T	E3
22	CS22	FOX, AMC, IPM, AMX, AMP, FEP, ERY, CTR, CIP	C8	E4
23	CS23	FOX, SPT, AMC, AMX, AMP, FEP, CST, CHL, ERY, CTR, CIP	C1, C2, C9	E4
24	CS24	FOX, AMC, IPM, AMX, AMP, FEP, ERY, CTR, CIP	C1, C2, C9	E4
25	CS25	FOX, SPT, AMC, AMX, AMP, FEP, CST, CHL, ERY, CTR, CIP	C1, O, C9	E4

^a^ NAL, nalidixic acid; FOX, cefoxitin; AZM, azithromycin; SPT, streptomycin; AMC, amoxicillin-clavulanic acid; IPM, imipenem; KAN, kanamycin; AMX, amoxicillin; NOR, norfloxacin; AMP, ampicillin; TET, tetracycline; FEP, cefepime; GEN, gentamicin; CST, colistin; CHL, chloramphenicol; ERY, erythromycin; CTR, ceftriaxone; CIP, ciprofloxacin. ^b^ S, *bla*_SHV_; O, *bla*_OXA_; T, *bla*_TEM_; C1, *bla*_CTX-M-1_; C2, *bla*_CTX-M-2_; C8, *bla*_CTX-M-8_; C9, *bla*_CTX-M-9._

## Data Availability

Not applicable.

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
