# Peer review of "Antibiotic Resistance and Molecular Characterization of Cronobacter sakazakii Strains Isolated from Powdered Infant Formula Milk"

_foods, 2022, doi:10.3390/foods11081093_

Round 1

Reviewer 1 Report

The submitted manuscript describes and discusses the results of an original research project carried out to antibiotic resistance and molecular characterization of Cronobacter sakazakii strains isolated from powdered infant formula milk. The manuscript describes and discusses logically designed experiments and presents results that are expected to be of large interest for the scientific community. It is an interesting study with an interesting approach. The paper in the whole is well designed and results sound. Nevertheless, the manuscript needs a minor revision:

  • In the introduction part should be more highlighted the main aim of the paper, and additionally, what is the novelty of carried research work.
  • How do the Authors select the analytes? The rational of the choice of the selected biologically active compounds studied is missing and should be clearly discussed.
  • Quality of the figures must be improved.

Author Response

Response to reviewer 1,

Dear reviewer 1

  • The introduction section is revised according to your comment. The main aim and novelty of the study is improved and highlighted in the text.
  • We selected the samples according to the sample size formula as previously described by other researchers. This statement is added and highlighted in the text.
  • The quality of all figures has been improved in the manuscript.

Kind regards,

Reviewer 2 Report

The research article "Antibiotic resistance and molecular characterization of Cronobacter sakazakii strains isolated from powdered infant formula milk" by Pakbin & colleagues, deals with an interesting topic. This is a well-written article & It is easy to follow the methods and results. However, I would suggest authors to improve discussion section.

Please pay extra attention to the naming rules of the bacterial species throughout the text. overall the manuscript is good.

Author Response

Dear reviewer 2

  • The discussion section is improved and highlighted in the text.
  • The naming rule of the bacterial species have been revised throughout the text (for example revised to the italics form) and highlighted in the text.

Kind regards
